# Shaping the Treatment Paradigm Based on the Current Understanding of the Pathobiology of Multiple Myeloma: An Overview

**DOI:** 10.3390/cancers12113488

**Published:** 2020-11-23

**Authors:** Slavisa Ninkovic, Hang Quach

**Affiliations:** 1Department of Haematology, St. Vincent’s Hospital Melbourne, Fitzroy, VIC 3065, Australia; slavisa.ninkovic@svha.org.au; 2Faculty of Medicine, University of Melbourne, Fitzroy, VIC 3065, Australia

**Keywords:** multiple myeloma, genomic aberrations, tumour microenvironment, minimal residual disease

## Abstract

**Simple Summary:**

Advances in our understanding of the causes of multiple myeloma and the mechanisms of disease progression have facilitated significant changes in the way we treat this incurable blood cancer. This review summarizes the genetic changes critical in disease development, the dysregulation of the immune system and how cells of the tumour microenvironment interact with and promote the survival of malignant plasma cells. We then provide a comprehensive overview of recent practice changing clinical trials incorporating novel agents into upfront treatment regimens as well as explore innovative immunotherapies, designed to overcome the dysfunctional immune system, and small molecules with novel mechanisms of action that are currently actively investigated in the relapsed and/or refractory disease setting.

**Abstract:**

Multiple myeloma is an incurable malignancy which despite progressive improvements in overall survival over the last decade remains characterised by recurrent relapse with progressively shorter duration of response and treatment-free intervals with each subsequent treatment. Efforts to unravel the complex and heterogeneous genomic alterations, the marked dysregulation of the immune system and the multifarious interplay between malignant plasma cells and those of the tumour microenvironment have not only led to improved understanding of myelomagenesis and disease progression but have facilitated the rapid development of novel therapeutics including immunotherapies and small molecules bringing us a step closer to therapies that no doubt will extend survival. Novel therapeutic combinations both in the upfront and relapsed setting as well as novel methods to assess response and guide management are rapidly transforming the management of myeloma.

## 1. Advances on the Pathobiology of Myeloma

Pathobiology of myeloma is multifaceted. Genomic aberrations ranging from gene mutations to large scale structural chromosomal anomalies contribute to vast clonal heterogeneity. Coupled with dysregulation of both the immune and non-immune tumour microenvironment (TME), these are considered the pillars critical to the development and progression of myeloma (Figure 1). 

### 1.1. Genomic Aberrations in MM

Modern techniques assessing whole exome, whole genome, RNA and single cell sequencing have progressed our understanding of the myeloma genomic landscape (Figure 1A). Little is known about genetic predisposition, however familial clustering of cases and a long-recognised racial predisposition suggest inherited risk. Recent genome wide association studies have identified recurrent single nucleotide polymorphisms localising to genomic regions robustly associated with myeloma [1,2]. Detectable primary genetic events, even in the pre-cursor condition of monoclonal gammopathy of undetermined significance (MGUS) are characterised by hyperdiploidy or translocations involving the *IGH* gene and somatic mutations in various oncogenes and tumour suppressor genes (TSG). Acquisition of secondary abnormalities characterised by translocations involving *MYC*, copy number variations and further somatic mutations predominantly in the MAPK, NFκB and DNA repair pathways contribute to vast clonal heterogeneity.

#### 1.1.1. Hyperdiploidy

Hyperdiploid (HRD) myeloma cells have between 48 and 75 chromosomes and are characterised by gain of the odd-numbered chromosomes 3, 5, 7, 9, 11, 15, 19 and 21 [3]. In general, hyperdiploidy is associated with a favourable outcome where trisomy 3 and 5 are linked to improved OS and trisomy 21 predicts poorer OS while co-existence of HRD with known adverse risk abnormalities (e.g., 17p deletion or t(4;14)) does not overcome their poor risk [4,5,6]. Gene expression profiling of HRD tumours reveals overexpression of genes associated with the MYC, NFkB and MAPK signalling pathways. 

#### 1.1.2. Chromosomal Translocations

Chromosomal translocations are the result of abnormal class switch recombination events mostly involving the *IGH* locus on chromosome 14, placing a partner oncogene under the control of the IGH promoter. The most common and prognostically significant translocations are summarised in Table 1 and include t(4;14)/*IGH-MMSET/FGF3*, t(11;14)/*IGH-CCND1*, t(14;16)*/IGH-MAF*, and t(14;20)/*IGH-MAFB* [7]. While bortezomib-based induction improves the outcome of patients with t(4;14), translocations involving *MMSET/FGF3 and MAF* remain high risk abnormalities even in the era of novel therapeutics [8,9,10]. It remains unclear however whether presence of t(14;16) alone, or the concomitant occurrence of other high risk abnormalities, seen in >80% of cases, results in worse outcomes [11]. The most common rearrangement, t(11;14), leading to upregulation of Cyclin D1 is a standard-risk cytogenetic aberration. Nonetheless, some clinical heterogeneity exists, as almost 50% of patients with plasma cell leukaemia, the most aggressive variant of myeloma, carry this translocation [12,13,14]. Plasma cells with t(11;14) show increased expression of BCL2, an anti-apoptotic protein and hallmark of tumour survival. Venetoclax, a first in class BCL-2 inhibitor, used in patients with t(11;14) has demonstrated marked efficacy, altering the prognostic implications of t(11;14) [15,16]. Unlike the simple reciprocal translocations involving *IGH*, *MYC* is dysregulated by complex translocations and insertions with multiple partner genes [17]. In general, *MYC* translocations are associated with poor outcome with *MYC* upregulation a marker of resistance to immunomodulatory drugs (IMiDs) [18]. 

#### 1.1.3. Copy Number Variations

Copy number variations (CNV) are generally secondary events characterised by gains or losses of either focal areas or an entire chromosome arm. The most common recurrent CNVs and commonly affected oncogenes and TSGs are presented in Table 1 [19]. The incidence of 1q rises with relapsed disease and presence of at least three copies with a minimum 20% clone size is indicative of relative resistance to bortezomib [20]. In a recent study, the presence of 4 or more copies of 1q or co-existence of other high risk abnormalities was associated with significantly poorer progression free survival even in the context of novel induction therapy with bortezomib-lenalidomide-dexamethasone [21]. The two main regions of interest in the short arm of chromosome 1 both contain TSGs, loss of which is associated with poor prognosis [22]. The prognostic implication of del(13q), which often co-exists with other high-risk aberrations, has a differential effect where monosomy 13 is adverse while partial deletion of the q arm, independent of other prognostic factors, appears protective [23]. Deletion of 17p is seen in 10% of cases at diagnosis with increasing frequency in the relapsed/refractory setting and *TP53* is the main deregulated TSG responsible for the adverse outcomes seen [24,25]. The size of the clone carrying the del(17p) is of significance with a cut-off of 60% and 20% portending significantly inferior outcomes in patients with low- and high-risk MM respectively [26]. 

While we rely heavily on cytogenetics to define high risk disease, the current classification is restrictive and over-simplified with heterogeneity in survival even among patients with t(4;14) or del(17p). In an attempt to improve risk classification Perrot and colleagues designed and validated a cytogenetic prognostic index with differing weight attributed to most clinically relevant abnormalities [27]. Using this innovative approach, patients with del(17p) as the sole abnormality or various combinations of adverse cytogenetic lesions were classed as high risk, meaning <35% of patients who displayed a t(4;14) were actually identified as high risk. Application of this prognostic index was validated externally and its discriminatory ability was deemed better than other combinations of traditional risk scores, including the R-ISS. Future real-world data set validation and potential application in design of risk-adapted treatment strategies are anticipated. 

#### 1.1.4. Somatic Mutations

Next generation sequencing (NGS) has facilitated the most significant advances in the understanding of the aberrant genomic landscape in myeloma [12,28,29]. These studies identified multiple recurrently mutated genes that may be of significance including *IRF4, KRAS, NRAS, MAX, HIST1H1E, RB1, EGR1, TP53, TRAF3, FAM46C, DIS3, BRAF, LTB, CYLD,* and *FGFR3*. Many of these lead to the deregulation of common signalling pathways with mutations in the MAPK and the non-canonical NFκB pathways dominating. It is not uncommon to simultaneously acquire mutations in multiple genes within the same pathway. Mutations are often subclonal and clonal heterogeneity is present almost universally even at diagnosis [28]. Clonal evolution may be characterised either by linear accumulation of aberrations or be branching in nature with several clonal progenitors present at diagnosis. Their dominance may alternate over time under selection pressures exerted by the microenvironment and therapy [30,31,32]. Of interest, the depth of response to initial induction therapy appears to be associated with the evolutionary patterns at time of relapse. Patients achieving complete response display a branching pattern of evolution with greater mutational burden, an altered mutational profile and often bi-allelic inactivation of TSGs and acquisition of structural abnormalities while those achieving a partial response have similar mutational profile at time of relapse as they did at diagnosis [33].

The clinical implications of NGS-derived data may have potential relevance in prognostic risk stratification, response assessment and detection of minimal residual disease and most excitingly, development of personalised and targeted therapeutics. In the Myeloma XI study, Walker and colleagues assessed the prognostic significance and correlated mutational burden to outcome in a group of newly diagnosed patients [12,34]. Mutations in the RAS and NFκB pathways appear prognostically neutral, *IRF4* and *EGR1* mutations may be protective and associated with response to lenalidomide while mutations in *CCND1* and the DNA repair pathway have a negative impact on OS [12,35]. While druggable mutations are seen in over 50% of patients, application of targeted therapy to selectively inhibit mutated genes is not without challenges. As the constantly evolving genomic landscape in myeloma is heterogenous, single agent therapy may suppress one clone but facilitate the emergence of others. Furthermore, while oncogenes may be activated, RNA sequencing has demonstrated that mutated genes may have low or undetectable expression, making targeted therapy futile [36].

### 1.2. The Immune Tumour Microenvironment

Immunoediting is a dynamic process between the immune system, the TME and the tumour itself. The initial phase is characterised by a dominant anti-tumour milieu and enhanced cytotoxic T- and natural killer (NK)-cell phagocytic activity, while decreased tumour immunogenicity, impaired immune surveillance and reduced anti-tumour activity through dysregulation of the T- and NK cell compartments, upregulation of inhibitory surface ligands and recruitment of immunosuppressive cells are the hallmarks of immune evasion [37]. During immune escape, a pro-tumour TME provides survival and growth signals to the tumour and leads to disease progression and resistance to therapy (Figure 1B). 

#### 1.2.1. T-cell Dysregulation in Myeloma

Qualitative and quantitative alteration of T-cells in both the peripheral blood and bone marrow, contributes to disease progression from as early as the MGUS stage [38]. Progressive reduction in the CD4/CD8 ratio due to reduction in helper T-cells and a relative increase in cytotoxic T-cells is independently associated with poor prognosis [39,40]. Active immune surveillance and expansion of cytotoxic T-cells is more prominent in MGUS and T-cells isolated from the bone marrow (BM) of MGUS patients exert vigorous anti-tumor specific responses to autologous malignant cells. This ability to control tumour progression is diminished in myeloma patients [39,41,42]. Indeed, patients with an expanded and functioning cytotoxic T-cell population have improved long-term disease control [43]. Cytotoxic T-cell dysfunction is best characterised by a combination of anergy, senescence and exhaustion. Malignant plasma cells induce T-cell anergy by presenting tumour antigens without co-receptor expression while repetitive antigen stimulation leads to reduced T-cell co-stimulatory receptor expression and senescent T-cells with impaired function and low proliferative capacity [44,45]. Persistent overstimulation and chronic antigen exposure finally lead to T-cell exhaustion characterised by expression of multiple inhibitory checkpoint receptors including programmed death receptor-1 (PD-1), T-cell immunoglobulin 3 (TIM-3) and lymphocyte activation gene-3 (LAG-3) which can be targeted with monoclonal antibodies (mAbs) to reinvigorate the T cells [45,46,47,48]. 

Helper T-cells are characterised by distinct cytokine secretion patterns and the entire repertoire is dysregulated in myeloma [38]. In newly diagnosed MM patients, a shift in the T_H_1/ T_H_2 cytokine profile creates a less cytotoxic milieu with lower levels of IFN-γ observed. Patients with higher proportion of T_H_1-cells tend to have more indolent disease and the T_H_2 phenotype tends to predominate in advanced stages of myeloma [41,44,49,50]. While the overall CD4 T-cell pool is reduced, the immunosuppressive regulatory T cell (T_reg_) component is increased and functionally competent in patients with MGUS and myeloma suggestive of a role in both pathogenesis and disease progression [51,52,53]. An intricate relationship also exists between T_regs_ and the pro-inflammatory T_H_17 cells with T_reg_-derived TGF-β promoting skewing of T_reg_/T_H_17 ratio further facilitating an immunosuppressive microenvironment [50,54]. 

#### 1.2.2. NK-cell Dysregulation in Myeloma

There are conflicting reports on the quantitative changes in NK cells although higher levels of circulating cytotoxic NK cells are seen in patients with long term disease control following stem cell transplant [38,55]. NK cell function can however be manipulated by malignant plasma cells and the surrounding TME. For example, NK cells isolated from the bone marrow of patients with MM demonstrate cytotoxic activity against myeloma cell lines yet have minimal effect on fresh autologous or allogeneic myeloma cells suggesting that ligands and/or cytokines from myeloma cells modulate the response. MICA (MHC class I polypeptide-related sequence A) expressed on the surface of plasma cells normally acts a ligand for the activating NK cell receptor, NKG2D, but shedding of MICA and concurrent downregulation of NKG2D on NK cells, leads to impaired cytotoxicity [56,57]. Other immunoevasive strategies employed by malignant plasma cells is upregulation of PD-L1 with concurrent upregulation of inhibitory receptor, for example PD-1, on NK cells with subsequent PD-1/PD-L1 interactions promoting NK cell functional exhaustion [58]. 

### 1.3. The Non-Immune Tumour Microenvironment

The interplay between plasma cells and various cellular and non-cellular components of the non-immune TME is similarly critical for the development and progression of myeloma [59]. Bone marrow stromal cells (BMSCs) are integral with BMSC-derived CXC-chemokine ligand 12 promoting the homing of plasma cells to the BM [60]. Physical interactions between BMSCs and plasma cells activates NF-κB and MAPK1 signalling pathways and promotes secretion of cytokines favouring plasma cell proliferation, inhibition of apoptosis and resistance to therapy [61,62]. BMSCs additionally secrete receptor activator of nuclear factor-κB ligand (RANKL) promoting osteoclast maturation. In conjunction with reduced levels of osteoprotegrin, a decoy RANK receptor, and the ability of other haemopoietic cells to differentiate into osteoclasts, there is net increase in the number and activity of osteoclasts culminating in myeloma related bone disease [63,64]. Other elements within the TME including endothelial cells which facilitate neo-angiogenesis or fibroblasts which differentiate to produce ECM components like fibronectin, which has been shown to confer drug resistance, exemplify how critical the microenvironment is for disease progression and for potential therapeutic targets being explored (Figure 1C) [59,65].

## 2. Advances on the Treatment of Multiple Myeloma

Over the last decade, improvement in survival for patients with MM boils down to three main reasons. First, earlier initiation of treatment, owing to the updated diagnostic criteria of MM which incorporates 3 specific biomarkers (clonal bone marrow plasma cells ≥60%, serum free light chain (FLC) ratio ≥100 provided involved FLC level is ≥100 mg/L, or ≥2 focal lesions on MRI) [66]. Second, efficacious upfront treatment with triplet therapies that incorporate proteasome inhibitors (PIs) and IMiDs, and now, the move to quadruplet therapy with the addition of anti-CD38 mAb upfront which will undoubtedly transform the outcomes for patients with myeloma [67,68,69]. Additionally, the development of various immunotherapies including bi-specific antibodies (bsAbs), antibody-drug conjugates (ADC), and chimeric antigen receptor (CAR) T-cells as well as small molecules with novel mechanisms of action have not only expanded the therapeutic armamentarium available at time of relapse, but have the capacity to drive the disease to an unprecedented depth of response. Third, the ability to detect minimal residual disease (MRD) at a sensitivity level of one tumour cells within 10^5^ or 10^6^ have paved way for a deeper response target that in turn, correlates to improved survival [70]. 

Indeed, a meta-analysis of studies assessing MRD status demonstrated that attaining MRD_neg_ was associated with improved progression free survival (PFS) (HR 0.35, *p* < 0.001) and OS (HR 0.48, *p* < 0.001) [71,72]. Increasingly, MRD assessment is therefore incorporated into clinical trials, given the potential of MRD_neg_ to act as a surrogate endpoint for survival outcomes for regulatory purposes. As there are several methods to assess MRD status, there is indeed a need to identify the most applicable, standardize its use and define the most ideal threshold for negativity [73,74]. Furthermore, the utility of ^18^fluorodeoxyglucose (^18^F-FDG) PET to identify foci of active disease not represented by the sample obtained for MRD assessment due to either extramedullary disease or heterogeneous marrow involvement, as well as novel methods to assess peripheral blood cell-free tumour DNA and investigate minute amounts of paraprotein by mass spectrometry are likely to complement current MRD techniques, further refining response criteria and disease monitoring [75,76,77]. 

### 2.1. Treatment of the Newly Diagnosed MM Patient

Maximising first-line therapy remains the best opportunity to optimise long term patient outcomes. In the current era of effective IMiDs and PI induction therapy, the incorporation of autologous stem cell transplant (AutoSCT) as part of upfront treatment remains the accepted standard of care for fit patients, as this has been shown to improve PFS compared to a non-AutoSCT approach [78,79]. While most clinical studies incorporating upfront AutoSCT only enrol patients ≤65 years of age, in many jurisdictions, patients up to the age of 70–75 are actively considered for an AutoSCT. Based on a thorough assessment of frailty, performance status and comorbidities and in turn improved patient selection and supportive care measures, there is sufficient evidence that AutoSCT is both safe and effective in patients up to the age to 75 [80,81]. These transplant-eligible (TE) patients usually have induction therapy consisting of 4–6 cycles of triplet combination of an IMiD, PI (or an alkylating agent) with corticosteroid, followed by AutoSCT with/without consolidation, then maintenance. For transplant ineligible (TNE) patients, induction with triplet or doublet therapy usually consists of an IMiD and/or PI with corticosteroids. Indeed, the recent addition of an anti-CD38 mAb to the IMiD and/or PI backbones for both TE and TNE patients has resulted in unprecedented depth and duration of response. Table 2 summarizes some of recent studies guiding the management of NDMM patients. 

#### 2.1.1. Induction Therapy

##### TE-NDMM

The ideal induction regimen for TE patients should rapidly reduce tumour burden to allow patients to proceed promptly to transplant without significant toxicities. Presently in most jurisdictions, three-drug combinations that incorporate bortezomib and/or lenalidomide is the current accepted standard of care for induction prior to ASCT, although thalidomide, a first generation IMiD is an acceptable alternative and remains standard of care in many European countries. While upfront bortezomib-lenalidomide-dexamethasone (VRd) induction with or without AutoSCT leads to impressive overall response rates (ORR) approaching 100%, indeed the depth and duration of response, including rates of MRD_neg_ and PFS are improved with transplantation (50 vs. 36 months, HR 0.65) [78,82]. Further support for upfront transplant as ongoing standard of care in the era of novel agents results from the EMN02/HO95 study which demonstrated improved PFS with AutoSCT as opposed to bortezomib-melphalan-prednisolone (VMP) intensification (mPFS 56.7 vs. 41.9 months, HR 0.73, *p* = 0.0001) [79]. In an attempt to improve on VRd induction, the ENDURANCE study assessed the substitution of bortezomib with the second-generation PI, carfilzomib (KRd) in a group of patients with no immediate intention to proceed to transplant. Preliminary results showed no significant difference in PFS (HR 0.98, *p* = 0.93), despite a higher rate of VGPR (73.8 vs. 64.7%) with KRd compared to VRd [83]. Note that this result pertains only to a group of non-high risk patients in whom a longer follow up is likely required for an appreciable benefit to show. The FORTE study, on the other hand, included patients with high risk cytogenetics. This study randomised TE patients, age ≤ 65 years, to either carfilzomib-cyclophosphamide-dexamethasone (KCd) or KRd followed by AutoSCT or KRd without ASCT. KRd was superior to KCd in inducing deep responses, even in high-risk patients with almost 50% achieving sustained MRD negativity and >80% remaining negative at 12 months [90,91]. In addition to sustained MRD_neg_, the addition of ASCT to KRd reduced the risk of early relapse in high risk patients, thus cementing the role of AutoSCT as part of upfront treatment for patients with MM [91]. 

The incorporation of the antiCD38 mAb, daratumumab (Dara) or isatuximab (Isa) to an IMiD and PI backbone will likely transform the outcomes for TE patients. Already, quadruplet combination of daratumumab-bortezomib-thalidomide-dexamethasone (Dara-VTd) is approved for upfront induction therapy in TE patients in some jurisdictions, based on the CASSIOPEIA study which demonstrated a 53% reduction in the risk of progression or death from Dara-VTd compared to VTd [84]. Impressive preliminary results follow from the GRIFFIN (Dara-VRd vs. VRd), MASTER (Dara-KRd), and GMMG-CONCEPT (Isa-KRd) studies which showed significantly improved depth of response in the anti-CD38 mAb-containing arm, and unprecedented rates of MRD_neg_ in the order of 30–80% [69,84,85,86]. Longer term follow-up of these studies is eagerly anticipated given at this stage we have no clear evidence of what impact the upfront addition of anti-CD38 mAbs has on PFS2 and overall survival. Notably, unlike the anti-CD38 mAbs, the addition of the mAb against SLAMF-7, elotuzumab, to VRd (Elo-VRd) did not improve depth of response post induction therapy, based on preliminary report of the GMMG-HD6 study although follow up is short and PFS data is awaited [92]. Similarly, Elo-VRd has not shown to improve outcome compared to VRd alone in a group of patients with newly diagnosed high risk MM according to the SWOG-1211 study [87]. 

##### TNE-NDMM

The TNE cohort of patients is heterogeneous with variable fitness levels requiring treatment to be based on thorough assessment of frailty and therapeutic goals. The IMW frailty score predicts mortality and the risk of toxicity in elderly patients, and while somewhat involved, it provides relevant prognostic information [93]. An abbreviated version based on 3 parameters, age, the Charlson Comorbidity Index, and the ECOG performance status is a suitable alternative validated by the FIRST study [88,94]. For TNE patients, while deep responses particularly MRD_neg_ is an important correlate to better survival outcome, it should not come at the expense of toxicity as early treatment discontinuation or ≥ grade 3 treatment emergent adverse events have been shown to independently correlate with increased mortality [74,88,95,96].

Induction with lenalidomide and dexamethasone (Rd) is a globally accepted standard of care with continuous therapy until disease progression associated with high response rates (81%, including 22% CR) and an mPFS of 26 months [88]. The addition of bortezomib to Rd in a cohort of patients not intending to immediately proceed to AutoSCT, was assessed in the SWOG S0777 study of 460 patients [89]. VRd demonstrated improved PFS and OS compared to Rd (41 vs. 29 months; *p* = 0.003 and not reached vs. 69 months; *p* = 0.0114, respectively), prompting the regulatory approval of VRd for the induction therapy for TNE patients in the US, Europe and Australia, and other jurisdictions. Notwithstanding, it is important to note that the patient population in the SWOG SO777 study was not truly representative of a TNE cohort; the majority of patients (57%) were <65 years of age and would otherwise be deemed TE. In multivariate regression analysis however the impact of VRd was retained irrespective of age and intent for AutoSCT. Grade ≥3 peripheral neuropathy occurred in approximately one third of patients which is likely contributed to by the intravenous route of administration and twice weekly dosing-schedule of bortezomib. An application of a modified version of VRD, so-called VRd-lite, that utilised a weekly schedule and subcutaneous administration of bortezomib with a lower dose of lenalidomide (15mg as opposed to 25mg) appeared to be well tolerated and efficacious in a small study of 53 TNE patients [97]. In many jurisdiction of the world that cannot access concurrent bortezomib and lenalidomide combinations, doublet or triplet therapy of Rd, VMP (bortezomib melphalan prednisone) or VCD (bortezomib cyclophosphamide and dexamethasone) remains acceptable standards of care (SOC) [98,99,100]. More recently, the addition of daratumumab to these backbones have resulted in a new SOC. Remarkable benefits are seen with the addition of daratumumab to Rd (Dara-Rd; MAIA study) with 92.9% ORR including 47.6% ≥CR, 24.2% MRD_neg_ and a significant improvement in mPFS compared to Rd (not reached vs. 31.9months, HR 0.56, *p* < 0.001) [67]. Equally, the addition of daratumumab to VMP (Dara-VMP; ALCYONE) also demonstrated significant improvements in depth and duration of response and critically overall survival [68,101]. Both Dara-Rd and Dara-VMP are approved in US and Europe for the initial treatment of TNE patients, with other jurisdictions hopefully following suit in the near future.

#### 2.1.2. Consolidation and/or Maintenance

Consolidation therapy refers to a short course of treatment, similar to that of induction, that is given post autoSCT with the aim of deepening initial response, whereas maintenance therapy refers to gentle intensity treatment that is given post initial phase of treatment for a prolonged period of time, if not until PD, with the aim to maintain disease response. To date, there has been insufficient data to determine whether consolidation therapy per se improves long term survival; several studies that incorporate consolidation post AutoSCT demonstrate improvement in depths of response and PFS but none of which have shown OS benefits [102,103]. The phase III StaMINA trial was perhaps the only study to assess the role of consolidation post AutoSCT in isolation. Here, patients aged <71 years (*n* = 758) were randomised after induction, 1:1:1 to either single AutoSCT, or consolidation with either a second (tandem) AutoSCT or 4 cycles of VRd, prior to maintenance lenalidomide therapy. No PFS or OS benefit was demonstrated by the consolidative approach with either VRd or second AutoSCT [104]. Indeed there may be little incremental benefit with consolidative therapy in the era of effective induction and maintenance therapy with novel agents, it is noted that in this trial, up to 12 months of induction therapy was allowed prior to randomisation, thus potentially providing a consolidative effect prior to study enrolment. Furthermore, to date there has been no sub analysis assessing for relative benefit in high risk patients who failed to achieve CR post AutoSCT. While further investigation is required, it would seem appropriate to design trials assessing MRD-driven consolidation strategies to determine the most appropriate combination and duration of any consolidation therapy.

A comparison of single versus tandem AutoSCT demonstrated non-inferiority of single transplant approach with no difference in the 2-year event free or overall survival and highlighted a high rate (26%) of patient refusal on non-medical grounds to proceed with a second transplant [105]. The StaMINA study, in the era of novel drugs, similarly reported non-inferiority of a single transplant with high drop-out rates while the EMN02/HO95 study demonstrated that depth of response is improved in patients with tandem transplant with more than 50% achieving a CR, leading to a 30% reduction in risk of death or progression [104,106,107]. The benefits are most pronounced in patients with high risk disease and the role of tandem AutoSCT should be discussed with these patients. 

Unlike consolidation, data for maintenance therapy is robust. Currently, lenalidomide monotherapy is the standard accepted maintenance with significant PFS and OS benefits demonstrated for both TE and TNE patients [88,108]. More recently, ixazomib, a second generation oral PI with a convenient once weekly schedule has emerged as a feasible alternative to lenalidomide maintenance in patients in whom lenalidomide is not an option. The TOURMALINE-MM3 and TOURMALIME-MM4 studies were pivotal phase III studies that compared ixazomib maintenance versus placebo in TE and TNE patients, respectively [109,110]. Ixazomib maintenance for 24 months post initial therapy in TE and TNE patients reduced the risk of progression or death by 38% and 36%, respectively but demonstrated no significant OS benefit. Ixazomib was very well tolerated with no cumulative side effects. Finally, continuous therapy with daratumumab, has also been explored in the TNE setting with evidence of improved depth of response both as monotherapy (ALCYONE), and in combination with lenalidomide (MAIA) [67,68]. 

### 2.2. Treatment of the Relapsed/Refractory MM Patient

At relapse specific considerations must be paid to patient age, frailty, comorbidities and the aggressiveness of disease recognizing that upon end-stage disease or in frail patients, the focus may need to be on symptom relief and the prevention of end-organ disease rather than attaining a deep and durable response. Indications for and the timing of initiation of therapy must be carefully considered given that asymptomatic biochemical relapse may have a slow tempo for a long period in some patients and may be monitored in the first instance. In contrast, patients with aggressive relapse or high risk disease are likely to do poorly and earlier treatment should be considered [111]. 

Knowledge of prior treatment history, depth and duration of response and the persistence and significance of treatment related toxicities is critical in choosing a suitable treatment. Participation in clinical trials should be offered to all patients where available.

#### 2.2.1. Early Relapse

The widespread use of lenalidomide and/or bortezomib containing therapy upfront, coupled with lenalidomide maintenance until disease progression has fashioned a cohort of lenalidomide-refractory patients at first relapse. Thus, there is an increasing shift favouring lenalidomide-sparing salvage regimen, incorporating either second-generation PI and/or mAb at second line treatment (Table 3).

In the ENDEAVOR study, carfilzomib-dexamethasone (Kd) was shown to be superior to bortezomib-dexamethasone (Vd) for patients who have had 1 to 3 prior lines of treatment, with respect to ORR, mPFS (18.7 vs. 9.4 months, HR 0.53, *p* < 0.001) and importantly OS (47.6 vs. 40.0 months, HR 0.791, *p* = 0.001) [112,113]. The benefit was seen in both lenalidomide-exposed (mPFS 12.9 vs. 7.3 months) and -refractory (mPFS 8.6 vs. 6.6 months) patients with improved response rates if given at first relapse [113]. More convenient once-weekly carfilzomib dosing was explored in CHAMPION-1, a phase 1/2 dose-finding study where 70 mg/m^2^ was deemed the maximum tolerated dose [126]. This was followed by the A.R.R.O.W. study which compared weekly Kd (carfilzomib: 70 mg/m^2^ IV days 1,8,15 in a 28 day cycle) with twice-weekly (Carfilzomib 27 mg/m2 IV days 1,2,8,9,15,16 in a 28-day cycle). Here, the weekly Kd was superior to twice weekly Kd with respect to ORR (63% vs. 41%) and PFS (HR 0.69, *p* = 0.0029) [114]. However, such conclusion should not be extrapolated to the current standard dose of Kd with carfilzomib 56 mg/m^2^ that was explored in the ENDEAVOR study and which has not been directly compared to weekly Kd schedule. Rather, post hoc cross-trial comparison of A.R.R.O.W., CHAMPION-1 and ENDEAVOR suggests comparable efficacy and tolerability, supporting weekly Kd as an appropriate backbone, particularly when used in combination in early relapse [(ORR 69.9 vs. 72.4%; mPFS 12.1 vs. 14.5 months; ≥ grade 3 AEs 67.6 vs. 85.3%)] [114,127,128]. 

The addition of mAbs to Vd or Kd has been assessed in several phase III studies (Table 3). At a median follow-up of 40 months, Dara-Vd (CASTOR; 76% R-exposed, 69% PI-exposed) was associated with prolonged PFS (16.7 vs. 7.1 months, HR 0.31, *p* < 0.0001) even in patients with high risk cytogenetics (12.6 vs. 6.2 months, HR 0.41, *p* = 0.0106) with most benefit seen at first relapse [115,116,129]. In the R–refractory cohort, PFS was short although Dara-Vd was superior to Vd (mPFS 7.8 months vs. 4,9m, p0.0002). In the phase III CANDOR study, Dara-Kd was compared with Kd, demonstrating marked improvement in PFS (not reached vs. 15.8 months, HR 0.63, *p* = 0.0027) with an impressive 53% risk reduction of disease progression in the R-refractory cohort [117]. Early reports on the phase III IKEMA study (Isatuximab-Kd vs. Kd) are similarly promising with 47% risk reduction of progression (mPFS not reached vs. 19.5 months; *p* = 0.007) [118]. 

Pomalidomide-based therapy was assessed in several studies including OPTIMISMM (VPd vs. Pd; mPFS 11.2 vs. 7.1 months, HR 0.61, *p* < 0.0001), ELOQUENT-3 (Elotuzumab-Pd vs. Pd; mPFS 10.3 vs. 4.7 months, HR 0.54, *p* = 0.008) and ICARIA (Isatuximab-Pd vs. Pd; mPFS 11.5 vs. 6.5 months, HR 0.60, *p* = 0.001). They all demonstrated significant improvement in PFS by addition of a PI or mAb to pomalidomide in a lenalidomide exposed and mostly refractory cohort of patients [119,120,121,130]. 

A lenalidomide backbone for RRMM remains a viable approach when it is combined with a novel agent, even for lenalidomide-exposed patients but especially for lenalidomide naïve patients. Addition of carfilzomib to Rd (KRd vs. Rd; ASPIRE) improved mPFS (26.3 vs 17.6 months, HR 0.69, *p* = 0.0001) with a difference of 9.3 months between lenalidomide naïve and exposed patients while lenalidomide refractory patients still derived benefit from KRd [122,131]. Ixazomib similarly prolonged PFS (20.6 vs 14.7 months, HR 0.74, *p* = 0.01; TOURMALINE-MM1) and was associated with a 26% reduction in risk of progression in lenalidomide-exposed patients [123]. Combination with mAbs appears tolerable with improvements in PFS with both elotuzumab (19.4 vs. 14.9 months, HR 0.70, *p* < 0.01; Elo-Rd vs. Rd; ELOQUENT-2) and daratumumab (45.8 vs. 17.5 months, HR 0.43, *p* < 0.0001; Dara-Rd vs. Rd; POLLUX) [124,125]. The greatest benefit of Dara-Rd is when given at first relapse (mPFS 53.3 months) suggesting incorporation of mAbs early in treatment of relapse is crucial [132]. 

#### 2.2.2. Later Relapse

The incorporation of IMIDs, PI and mAbs in the upfront therapy of MM and at early relapse invariably gives rise to heavily pre-treated populations at later relapses which are now termed triple (one IMiD, one PI and an anti-CD38 mAb), quad (as per triple with refractoriness to an additional IMiD or PI) or even penta (refractory to 2 IMiDs, 2 PIs, and an anti-CD38 mAb)-refractory [133]. Indeed patients refractory to an anti-CD38 mAb have a median OS of 9.3 months and penta-refractory patients have dismal outcomes with a median OS of 5.6 months [133]. Immunotherapies designed to reverse the qualitative and quantitative deficits and invigorate the immune system, as well as small molecules with novel mechanisms of action play a significant role in this setting. 

##### Novel Immunotherapies—ADCs, bsAbs, CAR T-cells

In addition to mAb against CD38 and SLAMF7, immunotherapies targeting B-cell maturation antigen (BCMA) have revolutionised treatment and offer new hope for patients with multiply refractory MM. BMCA is expressed universally on and associated with plasma cell proliferation, survival and development of drug resistance [134]. BCMA-targeted immunotherapies that are well into clinical development include BCMA-antibody drug conjugates (ADC), chimeric antigen receptor (CAR) T cells, and bispecific T or NK cell engagers. 

Belantamab mafodotin an antibody-drug conjugate (ADC), is a humanised IgG1 anti-BCMA antibody conjugated to a potent tubulin polymerisation inhibitor, monomethyl Auristatin F, which once internalised causes direct cellular apoptosis. Interaction with the Fc receptor on immune effector cells also promotes cytotoxicity while engagement of BCMA blocks signalling through this pathway [135,136]. In early phase studies belantamab mafodotin demonstrated impressive and durable single agent activity which has been confirmed in a phase II study (DREAMM-2; 31–34% ORR) [137,138]. Preliminary results of the DREAMM-6 study, of the arm containing belantamab mafodotin in combination with Vd in RRMM patients (median 3 prior lines) are promising with an ORR 78% and further studies assessing its efficacy in combination with other current standard anti-myeloma therapy are underway [139]. 

Bi-specific T-cell (or NK-cell) engagers are antibody constructs that simultaneously bind CD3 on endogenous T-cells (or CD16 on NK cells) and TAA on myeloma cells, bringing them into direct proximity. This creates an immunological synapse facilitating myeloma specific cytotoxicity in a process independent of co-stimulation or antigen presentation. The bsAbs differ in the construct itself as well as the target TAA with BCMA being the most common. A construct composed of two single-chain variable fragments and a short linker, known as BiTE compounds, results in a protein with a short half-life necessitating continuous infusion. Constructs of T cell engagers (TCE) with an Fc domain resembling an ‘IgG-like’ molecule, constitute a longer half-life, allowing for intermittent dosing. Pre-clinical studies with AMG420 and AMG701 constructs targeting BCMA demonstrated potent in vitro autologous lysis of myeloma cell isolated from ND and RR myeloma patients [140,141]. AMG420 was the first BCMA BiTE to show proof of concept of efficacy in heavily pre-treated patients with RRMM, demonstrating a response rate of 70% with 50% MRD negativity at the maximum tolerated dose [142]. This molecule is now no longer in development, in favour of the longer half-life construct, AMG701. More recently, preliminary results from studies of two other competing extended half-life BCMA-targeting TCE have shown remarkable efficacy. CC-93269 is a humanized TCE that binds to BCMA bivalently. Preliminary results from a phase I study of 30 patients, 67% of whom were penta-refractory, demonstrated an ORR of 43% (≥CR 16.7%) [143]. Notably, at the highest dose of 10mg, ORR was 88.9% (≥CR 44.4%). Time to response was rapid (median 4.1 weeks) and 12 out of 13 responding patients achieved MRD negativity before C4D1 within the bone marrow. Teclistamab, another humanized BCMA TCE, has similarly shown remarkable efficacy in preliminary report of a phase I study in 78 patients (80% triple-class refractory; 41% penta-drug refractory). Of the 12 patient who were treated at the highest dose of teclistamab, 270mcg/kg, ORR was 67% (≥VGPR 50%) with some patients achieving MRD negativity [144]. These results, hitherto, have been unprecedented in a group of heavily pre-treated patients and rival that of chimeric antigen receptor (CAR) T-cells.

CAR T-cells are autologous T-cells genetically modified ex vivo resulting in a novel T-cell receptor with novel antigen specificity, and an extracellular antigen binding domain bypassing the conventional MHC-dependent binding of T-cells and the intracellular signalling required for T-cell activation/co-stimulation. Table 4 summarizes recent updates on three major CAR T-cell studies all targeting BCMA [145,146,147]. Orvacabtagene autoleucel (orva-cel) is a BCMA CAR-T cell with low binding affinity for soluble BCMA and a manufacturing process that enriches for central memory T-cell phenotype, potentially increasing persistence and durability. In the phase 1/2 EVOLVE study (*n* = 62, 94% triple- and 48% penta-refractory), orva-cel was highly effective across all dose levels (ORR 92%, ≥VGPR 68%, 84% MRD_neg_ in evaluable patients) with limited follow-up and ongoing enrolment at the recommended phase 2 dose of 600 × 10^6^ CAR-T cells [137]. Idecabtagene vicleucel (ide-cel), a BCMA CAR-T product with a 4-1BB co-stimulatory domain to promote CAR-T cell survival and proliferation was assessed in the KarMMa study (*n* = 128, 84% triple- and 26% penta-refractory) [138]. Ide-cel in these heavily pre-treated patients (median 5 prior lines) demonstrated deep and durable responses (73% ORR, 94% MRD_neg_ in evaluable patients, 10.7 months median duration of response) with a median PFS of 8.8 months (20.2 months in patients achieving a CR) and a median OS of 19.4. In the CARTITUDE study, JNJ-4528, a dual epitope CAR containing two heavy chain variable fragments against different BCMA epitopes and a 4-1BB co-stimulatory domain was assessed in a similarly heavily pre-treated cohort of patients, 86% of whom were triple-refractory [139]. With an impressive 100% ORR and 86% sCR, 81% of evaluable patients also achieved MRD negativity and at a median follow-up of 11.5 months, the 9-month PFS rate is 86%. Compared to the dismal outcome penta-exposed and refractory patients face in general, these results are very promising. 

Notwithstanding the unprecedented efficacy, novel immunotherapies are not without significant challenges including unique dose-limiting toxicities and potentially serious complications including infusion related reactions (IRR), cytokine release syndrome (CRS) and immune effector cell associated neurotoxicity (ICANS). IRR with belantamab mafodotin is mild; in DREAMM-2 IRR were predominantly grade 1–2, occurring mostly during the first treatment [124]. Of more pertinence is corneal toxicity, which was the most common grade 3–4 adverse event (27% in the 2.5 mg/kg and 21% in the 3.4 mg/kg cohort), with changes to the corneal epithelium mediated by macropinocytosis of the MMAF cytotoxic payload in corneal cells. While the ocular toxicity is reversible, it is often dose-limiting with dose delays in almost 50% of patients and dose reductions in a quarter of patients, while 4 patients permanently discontinued treatment. Unlike ADCs, two toxicities of special interest with bsAbs/TCEs and CAR T-cells are CRS and ICANS. CRS occurs due to non-antigen specific immune activation and is associated with elevated levels of several cytokines including interleukin-6 (IL-6) driving the immune dysregulation. In mild cases it may present as a flu-like illness while grade 3 or 4 toxicity is associated with life-threatening cardiovascular, pulmonary and renal involvement. ICANS may occur during CRS or more commonly after CRS has subsided. It is usually characterised by a toxic encephalopathy with word finding difficulties, aphasia and confusion which may progress to depressed levels of consciousness, seizures and cerebral oedema in more severe cases. The exact pathophysiology is not completely understood but cytokines, chemokines and the degree of CAR T-cell expansion are proposed to play a role. Early recognition of CRS and ICANS and prompt intervention, ranging from supportive care to administration of anti-IL-6 mAb tocilizumab, is crucial for the prevention of catastrophic consequences. The incidence of CRS and ICANS in the above-mentioned CAR T studies is summarised in Table 4 with grade 3 or worse CRS seen in 3–7% of cases while grade 3 or worse ICANS was similarly not uncommon with 3% of patients experiencing significant neurological toxicity [137,138,139].

##### Small Molecules—Selinexor, Venetoclax and Iberdomide

In addition to immunotherapies, several small molecules including selinexor, venetoclax and iberdomide are transforming the treatment of RRMM. Selective inhibitor of nuclear export (SINE) compounds target exportin 1 (XPO1) a prominent nuclear exporter that controls the nuclear-cytoplasmic localisation of over 200 proteins [148,149]. Selinexor, a first-in-class SINE compound blocks XPO1, overexpressed in myeloma, and forces nuclear accumulation and activation of tumour suppressor proteins, inhibits NFkB and reduces oncoprotein mRNA translation [150,151]. Clinical trials investigating selinexor, both as single agent and in combination with current anti-myeloma therapies, show evidence of significant efficacy in patients with triple-refractory disease [152,153,154,155]. Combination therapy with Vd (SVd; BOSTON study) in patients with 1–3 prior therapies led to 76.4% ORR and a 30% reduction in risk of progression or death (mPFS 13.9 vs 9.4 months, *p* = 0.0075) [156]. 

Venetoclax is of particular interest in patients with t(11;14) with clinical evidence of single agent activity in the majority of heavily pre-treated patients carrying the translocation [157]. BELLINI, a study of venetoclax or placebo with Vd included 13% patients with t(11;14) and 79% patients with high levels of *BCL*2 expression by immunohistochemistry [158]. The addition of venetoclax to Vd (Ven-Vd) resulted in a favourable risk-benefit profile only in patients with t(11;14) or high levels of BCL-2. After a median follow-up of 22.7 months, patients with t(11;14) treated with Ven-Vd had not reached median PFS compared to 9.3 months for the placebo treated cohort. Furthermore, biomarker analysis demonstrates that both presence of t(11;14) and high levels of *BCL2* expression as assessed by quantitative PCR are associated with improved response of venetoclax [159]. 

Iberdomide, a small molecule explored in myeloma is a cereblon E3 ligase modulator (CELMoD), which binds cereblon with up to 20 times higher affinity than lenalidomide or pomalidomide causing degradation of ikaros and aiolos [160]. In preclinical models it has significant anti-myeloma and immunostimulatory activity with synergism when combined with bortezomib and daratumumab [161,162]. In a phase Ib/IIa dose-escalation study, iberdomide was assessed alone or in combination with dexamethasone and daratumumab or bortezomib [163]. The combination with dexamethasone in heavily pre-treated patients (median 5 prior lines; 86% refractory to both IMiDs) demonstrated efficacy with ORR 32.3%. Studies with CC-92480, a more potent CELMoD, are also in early stages with very promising single agent activity and 55% ORR at maximum tolerated dose in a cohort of patients where two thirds were penta-refractory [164]. 

## 3. Future Direction and Conclusions

The significant advances on the pathobiology and translation of this knowledge into novel therapeutics and the management of myeloma, is providing unprecedented outcomes for patients with this incurable malignancy. Although treating myeloma is becoming more satisfying, we as a community face several immediate challenges. While many novel immunotherapies are established, elucidating their most appropriate sequencing to preserve and reverse the dysfunctional host immune system will be as crucial as their incorporation into earlier treatment regimens. Mitigating dose-limiting and potentially life-threatening toxicities, furthering our understanding and potentially pre-empting the emergence of resistance and overcoming the vast financial toxicity associated with these therapeutics making them inaccessible for many patients around the world are some of the initial priorities. In the intermediate term, large scale international studies incorporating translational research aimed at unravelling the genomic landscape and dissecting the tumour microenvironment biology are imperative for the future treatment of myeloma. The development of novel prognostic biomarkers and incorporation into improved risk stratification models to better identify myeloma subtypes and risk groups will be critical. Advancement of highly sensitive, cost-effective, readily available and standardised MRD techniques will allow us to better assess response and potentially determine treatment intensity and duration as we move towards response-adapted strategies to therapy with both escalation and de-escalation options integrated into treatment algorithms. Finally, as we continue to make these advances and better understand the interplay between dominant aberrations for an individual patient at various stages of disease, and with the ever-expanding therapeutic armamentarium available, we approach the ultimate goal of personalised medicine and potentially a cure.

## Figures and Tables

**Figure 1 cancers-12-03488-f001:**
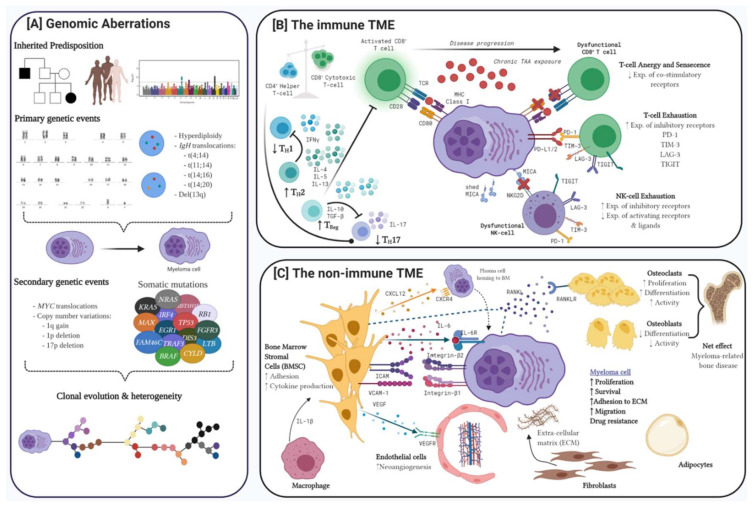
Myeloma pathogenesis is characterised by (**A**) genomic aberrations, dysregulation of (**B**) the immune tumour microenvironment (TME) and (**C**) the multifarious interplay between myeloma cells and the non-immune TME. (**A**) Familial clustering, racial predisposition and genome wide association studies highlight a familial predisposition for myeloma. Primary genetic events may occur before development of myeloma while secondary events and various somatic mutations lead to clonal evolution and heterogeneity and ultimately disease progression and resistance to therapy. (**B**) The dysregulated immune system is characterised by progressive increase in cytotoxic T and NK cells which through reduced expression of co-stimulatory receptors and increased expression of inhibitory receptors become dysfunctional. Helper T-cells in turn are diminished with a shift to more pro-inflammatory and immunosuppressive cytokine milieu exerted by T_H_2 and regulatory T cells. (**C**) The interaction between malignant plasma cells and various TME components, especially bone marrow suppressor cells, is critical in promoting myeloma cell survival and resistance to therapy. Created with BioRender.com.

**Table 1 cancers-12-03488-t001:** Summary and prognostic implications of common primary and secondary genomic aberrations seen in patients with multiple myeloma at diagnosis.

Aberration Category		Genomic Aberration	Involved Oncogenes and TSG	Frequency, %	Prognosis
**Primary genetic events**	**Hyperdiploidy**	Chromosomes 3, 5, 7, 9, 11, 15, 19 and 21 trisomy	Overexpression of MYC, NFκB and MAPK pathways	50	Favourable
	**Chromosomal translocations**	t(4;14)	*IGH-MMSET/FGF3*	10–15	Adverse
		t(11;14)	*IGH-CCND1*	15–20	Neutral
		t(14;16)	*IGH-MAF*	5	Adverse
		t(14;20)	*IGH-MAFB*	1	Adverse
**Secondary genetic events**	**Chromosomal translocations**	*MYC* translocations	Multiple partners incl. *IGH, IGL, IGK, FAM46C, CCND1, XBP1*	15–20	Neutral or Adverse
	**Copy number variations**	1q gain	*CKS1B* *ANP32E* *BCL-9*	35–40	Adverse
		1p deletion	*FAM46C* *FAF1* *CDKN2C*	30	Adverse
		13q deletion *	*RB1*	45–50	Neutral or Adverse
		17p deletion	*TP53*	10	Adverse

TSG = tumor suppressor gene; * Del(13q) is considered a primary CNV.

**Table 2 cancers-12-03488-t002:** Selected clinical studies guiding management of newly diagnosed transplant eligible and ineligible multiple myeloma patients.

Clinical Study			Response Rate, %				Med PFS, Months	Overall Survival
	Study Population	Regimen	ORR	CR	≥VGPR	MRD_neg_		
**Newly diagnosed multiple myeloma, transplant eligible (TE)**								
**IFM 2009** [78]	Phase III *n* = 700 Med age = 59y	VRd (x3) > VRd (x5) > R maint. vs. VRd (x3) > ASCT + VRd (x2) > R maint.	97 vs. 98	48 vs. 59 *p* = 0.03	77 vs. 88 *p* = 0.001	65 vs. 79 *p* < 0.001	36 vs. 50 HR 0.65 *p* < 0.001	4yr OS 82 vs. 81% HR =1.16 *p* = 0.87
**PETHEMA/GEM2012** [82]	Phase III *n* = 458 Med age = 58y	VRd (x6) > ASCT > VRd (x2)	81	44	75	45.2 (@10^−6^) 65.9 (@10^−4^)	NR	NR
**EMN02/HO95** [79]	Phase III *n* = 1197 Med age = 58y	VCD induction 1st randomisation to ASCT vs. VMP intensification [results reported here] 2nd randomisation to VRd vs. observation	95 vs. 95	44 vs. 40	84 vs. 77		56.7 vs. 41.9 HR = 0.73 *p* = 0.0001	5yr OS 75.1 vs. 71.6% HR = 0.90 *p* = 0.35
**ENDURANCE** [83]	NDMM * ***No immediate intention for ASCT*** *n* = 1087 Med age = 65y ** del(17p), t(14;16) & t(14;20) excluded*	VRd vs. KRd Followed by second randomisation to R maintenance until PD vs. 2 yrs.	84 vs. 87 *p* = 0.13	15 vs. 18 *p* = 0.261	65 vs. 74 *p* = 0.002		34.4 vs. 34.6 HR 1.04 *p* = 0.70	3yr OS 84 vs. 83% HR 0.98 *p* = 0.923
**FORTE** [84,85]	Phase III *n* = 474 Med age = 57y	(A) KCd > ASCT > KCd (*n* = 159) (B) KRd > ASCT > KRd (*n* = 158) (C) KRd (x12) (*n* = 157)		60 vs. 61	89 vs. 87	58 vs. 54	Data immature	Data immature
				*Comparisons are between cohort (B) KRd > ASCT > KRd and cohort (C) KRd (x12)*				
**CASSIOPEIA** [84]	Phase III *n* = 1085 Med age = 59y	VTd vs. Dara-VTd > ASCT > VTd or Dara-VTd consolidation > 2nd randomisation to Dara maintenance vs. observation until PD	*Response assessment at D100 post ASCT*					
			89.9 vs. 92.6 *p* = 0.11	26 vs. 39 *p* < 0.0001	78 vs. 83 *p* = 0.024	20 vs. 34 *p* < 0.0001	18m PFS 85 vs. 93% HR 0.43 *p* < 0.0001	Data immature
**GRIFFIN** [69]	Phase III *n* = 207 Med age = 60y	VRd (x4) vs. Dara-VRd > ASCT > x2 consolidation > 2nd randomisation to R (x26) vs. Dara-R maintenance	*Response assessment at end of consolidation*					
			92 vs. 98 *p* = 0.016	32 vs. 42.4 *p* = 0.068	73 vs. 91 *p* = 0.0014	16.5 vs. 47.1	NR at med f/u 22.1m	Data immature
**MASTER** [85]	Phase II *n* = 81 Med age = 61y	Dara-KRd (x4) > ASCT > Dara-KRd (MRD-directed x4-9) > R maintenance	*Response assessment at end of consolidation*					
			100	95	100	82% (@10^−5^)	Data immature	Data immature
**GMMG-CONCEPT** [86]	* **Both TE & TNE** Phase II High risk pts *n* = 153, interim report on *n* = 50	Isa-KRd TE: Isa-KRD (x6) > ASCT; *n* = 46 TNE: Isa-KRd (x8); *n* = 4	100	46	90	31 (20/33 pts)	Data immature	Data immature
**SWOG 1211** [87]	Phase II High risk pts *n* = 103 Med age = 62.4y	VRd (x8) vs. Elo-VRd(x8) > attenuated VRd maintenance	88 vs. 83	6 vs. 2.1			34 vs. 31 HR 0.97 *p* = 0.449	NR vs. 68 HR 1.28 *p* = 0.239
**Newly diagnosed multiple myeloma, transplant ineligible (TNE)**								
**FIRST** [88]	Phase III *n* = 1623	Rd continuous vs. Rd (18 months) vs. MPT	*Response assessment for Rd cohort only*					
			81	22	48		26	59.1
**SWOG S0777** [89]	***Both TE and TNE with no immediate intention for ASCT*** *n* = 460 Age ≥ 65 = 43%	Rd (x6) vs. VRd (x8) > R maintenance	78.8 vs 90.2	12.1 vs. 24.2	53.2 vs. 74.9		29 vs. 41 HR 0.74 *p* = 0.003	69 vs. NR HR 0.71 *p* = 0.0114
**ALCYONE** [68]	Phase III *n* = 706 Med age = 71y	VMP (x9) vs. Dara-VMP f/b Dara maintenance in Dara-VMP cohort	73.9 vs. 90.9 *p* < 0.0001	25 vs. 46 *p* < 0.0001	50 vs. 73 *p* < 0.0001	7 vs. 28 *p* < 0.0001	18.1 vs. NR *p* < 0.001	36m OS estimate 67.9 vs. 78% HR 0.60 *p* = 0.0003
**MAIA** [67]	Phase III *n* = 737 Med age = 73y	Rd vs. Dara-Rd	81.3 vs. 92.9 *p* < 0.001	24.9 vs. 47.6 *p* < 0.01	53.1 vs. 79.3 *p* < 0.01	7.3 vs. 24.2 *p* < 0.001	31.9 vs. NR HR 0.56 *p* < 0.001	Data immature

PFS = progression free survival; OS = overall survival; NDMM = newly diagnosed multiple myeloma; TE = transplant eligible; VRd = bortezomib, lenalidomide, dexamethasone; f/b = followed by; R = lenalidomide; ORR = overall response rate; CR = complete response; sCR= stringent CR; VGPR = very good partial response; MRD_neg_ = minimal residual disease negativity; HR = hazard ratio; NR = not reached; KRd = carfilzomib, lenalidomide, dexamethasone; KCd = carfilzomib, cyclophosphamide, dexamethasone; HRA = high risk cytogenetic abnormalities; PD = progressive disease; VTD = bortezomib, thalidomide, dexamethasone; Dara = Daratumumab; rand. = randomisation; induct. = induction; consol. = consolidation; maint. = maintenance; Elo = elotuzumab; Rd = lenalidomide, dexamethasone; MPT = melphalan, prednisolone, thalidomide; VMP = bortezomib, melphalan, prednisolone.

**Table 3 cancers-12-03488-t003:** Selected clinical studies guiding management of early relapse in MM.

Median Progression Free Survival (mPFS), Months									
	Study Population	Regimen	Med f/u (m)	Overall	R_naïve_	R_exp._	R_ref._	PI_exp._	PI_ref._
**Proteasome Inhibitor-based**									
**ENDEAVOR** [112,113]	Phase III, *n* = 929 Med # PL = 2 (1–2) Prior R = 38% Prior V = 54%	Kd vs. Vd	11.9	18.7 vs. 9.4 HR 0.53 *p* < 0.0001		12.9 vs. 7.3 HR 0.67	8.6 vs. 6.6 HR 0.80	15.6 vs. 8.1 HR 0.56	
**A.R.R.O.W.** [114]	Phase III, *n* = 478 Med # PL = 2–3 Prior R = 84% (74.5% ref.) Prior V = 98.9% (42% ref.)	Kd 70QW vs. Kd 27BIW	12.6	11.2 vs. 7.6 HR 0.69 *p* = 0.0029		HR 0.72	HR 0.76		HR 0.73
**CASTOR** [115,116]	Phase III, *n* = 498 Med # PL = 2 (1–10) Prior IMiD = 75.7% Prior PI = 68.5%	Dara-Vd vs. Vd	40	16.7 vs. 7.1 HR 0.31 *p* < 0.0001	16.7 (Dara-Vd)	9.5 (Dara-Vd)	7.8 (Dara-Vd)		
**CANDOR** [117]	Phase III, *n* = 466 Med # PL = 2 (1–2) Prior R = 42.2% (33% ref.) Prior V = 90.3% (29% ref.)	Dara-Kd vs. Kd	17	NR vs. 15.8 HR 0.63 *p* = 0.0027	HR 0.71	HR 0.53	HR 0.47		
**IKEMA** [118]	Phase III, *n* = 302 1–3 prior lines Prior R = 78% (33% ref.) Prior PI = 90% (33% ref.)	Isa-Kd vs. Kd	20.7	NR vs. 19.2 HR 0.53 *p* = 0.007					
**Pomalidomide-based**									
	**Study population**	**Regimen**	**Med f/u (m)**	**Overall**	**R_naïve_**	**R_exp._**	**R_ref._**	**PI_exp._**	**PI_ref._**
**OPTIMISMM** [119]	Phase III, *n* = 559 1–3 prior lines Prior R = 100% (70% ref.) Prior V = 72% (10% ref.)	VPd vs. Vd	15.9	11.2 vs. 7.1 HR 0.61 *p* < 0.0001		11.2	9.5		
**ELOQUENT-3** [120]	Phase III, *n* = 117 Med # PL = 3 (range 2–8) Prior R = 99% (87% ref.) Prior V = 100% (80% ref.)	Elo-Pd vs. Pd	9.1	10.3 vs. 4.7 HR 0.54 *p* = 0.008			10.2		
**ICARIA** [121]	Phase III, *n* = 307 Med # PL = 3 (range 2–4) Prior R = 100% (93% ref.) Prior V = 100% (76% ref.)	Isa-Pd vs. Pd	11.6	11.5 vs. 6.5 HR 0.60 *p* = 0.001					
**Lenalidomide-based**									
**ASPIRE** [122]	Phase III, *n* = 792 Med # PL = 2 (1–3) Prior R = 19.8% Prior V = 65.8%	KRd vs. Rd		26.3 vs. 17.6 HR 0.69 *p* = 0.0001	28.7 (KRd) *n* = 317	19.4 (KRd) *n* = 79	9.3 (KRd) *n* = 57		
**TOURMALINE-MM1** [123]	Phase III, *n*= 722 1–3 prior lines Prior R = 12% Prior V =69%	Ixa-Rd vs. Rd	14.7	20.6 vs. 14.7 HR 0.74 *p* = 0.01	NR vs. 17.5 HR 0.74			18.4 vs. 13.6 HR 0.74	
**ELOQUENT-2** [124]	Phase Ib-II, *n*= 646 Med # PL = 2 (1–4) Prior R = 6% Prior V = 70%	Elo-Rd vs. Rd	24.5	19.4 vs. 14.9 HR 0.70 *p* < 0.001					
**POLLUX** [125]	Phase III, *n* = 569 Med # PL = 1 (1–11) Prior R = 55.1% (3.7% ref.) Prior V = 86.1% (18.1% ref.)	Dara-Rd vs. Rd		45.8 vs. 17.5 HR 0.43 *p* < 0.0001		38.6 vs. 18.6 HR 0.34			

PL = prior lines; ref = refractory; exp. = exposed; KRd = carfilzomib, lenalidomide, dexamethasone; Rd = lenalidomide, dexamethasone; R = lenalidomide; Elo = elotuzumab; V = bortezomib; Kd = carfilzomib, dexamethasone; QW = once weekly; BIW = twice weekly; VPd = bortezomib, pomalidomide, dexamethasone; Pd = pomalidomide, dexamethasone.

**Table 4 cancers-12-03488-t004:** Summary of pivotal CAR-T studies targeting BCMA.

	EVOLVE [145]	KarMMa [146]	CARTITUDE-1 [147]
Product	Orvacabtagene autoleucel [orva-cel]	Idecabtagene vicleucel [ide-cel]	JNJ-4528
Target Antigen	BCMA	BCMA	BCMA
*n*	62	128	29
Median age (range)	61 (33–77)	61 (33–78)	60 (50–75)
High risk cytogenetics *, %	41	35	27
Median prior lines of therapy, (range)	6 (3–18)	6 (3–16)	5 (3–18)
Triple-refractory, %	94	84	86
Penta-refractory, %	48	26	31
Bridging chemotherapy required, %	63	88	79
Median follow-up time, months	6.9	13.3	9.0
ORR, %	92	73	100
sCR/CR, %	36	33	86
≥VGPR, %	68	53	97
MRD_neg_, %	84	94	81
Median duration of response, months	NR	10.7 [19.0 months in CR pts]	NR
Median PFS, months	NR mPFS (lowest dose cohort) = 9.3 m	8.8 (5.6–11.6) [20.2 months in CR pts]	NR 9 month PFS = 86%
Median OS, months	NR	19.4 (18.2-NR)	NR
Any grade CRS, %	89	84	93
≥ Grade 3 CRS, %	3	6	7
Median time to CRS, days (range)	2 (1–4)	1 (1–12)	7 (2–12)
Median duration of CRS, days (range)	4 (1–10)	5 (1–63)	4 (2–64)
Any grade ICANS, %	13	17	10
≥ Grade 3 ICANS, %	3	3	3

NR = not reached. * Presence of t(4;14), t914;16), del(17p), or 1q21 abnormality. BCMA = B-cell maturation antigen, CRS = cytokine release syndrome, ICANS = immune effector cell associated neurotoxicity syndrome.

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
