# Peer review of "Shaping the Treatment Paradigm Based on the Current Understanding of the Pathobiology of Multiple Myeloma: An Overview"

_cancers, 2020, doi:10.3390/cancers12113488_

Round 1

Reviewer 1 Report

The authors performed a very nice and interesting overview of the current scientific background concerning multiple myeloma.This review is well written and designed, coincise and reads well and provides a useful picture of nowadays multiple myeloma for readers.

Please find some minor comments to make the manuscript more informative and few corrections

1) sections 1.1.2 and 1.1.3, please add as references the work on t14;16 by Mina et al (BCJ 2020), on 1q by Schmidt et al (BCJ 2020) and on del17 by Perrot et al (JCO 2019).

2) Page 4 L 17: check Myeloma Xi -> XI

3) Section 2.1: When discussing ASCT and the eligibility criteria to define transplant eligibilty, I would add for clarity that there is enough evidence that ASCT is safe and effective even among patients up to the age of 75. In this light it would be useful citing the ASCO2019 educational on this topic (Mina R, Bringhen S, Wildes TM, Zweegman S, Rosko AE.)

4) TAble 2: please add data from the EMN02/HOVON65 study

5) Section 2.1.1. Please specificy that induction therapy in ASCT eligible patients includes a PI, mainly bortezomib and an IMid, lenalidomide but also thalidomide, as the latter has been and still is a SoC in many european countries.

6) In the upfront treatment section, I would add few words about the fact that we still don't know, for the most part, the impact on PFS2/OS of upfront use of anti-CD38, except for the alcyone study, in which however only 10-11% of patients received daratumumab in a subsequent line. Although its use as front line agent might translate into an OS advantage this still has to be clearly demonstrarted in both ASCT and nASCT settings.

7) Geriatric assessment: the authors state that the use of the IMWG frailty core might be cumbersome in the clinic; I suggest rephrasing this sentence, as with the online calculator it takes only 5 minutes to fill all the forms and provides important prognostic pieces of information that, at least in the elderly setting, might be as important as cytogenetics.

8) When citing the SWOG777 (VRd vs Rd) data, please report also PFS and OS among patients over 65 years of age.

9) Please briefly discuss the role of double ASCT as consolidation (EMN02, Stamina), which is a viable option among high risk patients.

10) Maintenance: Given that there is no formal randomized trial and that PFS data with ixazomib seem less strong than those with lenalidomide, also without OS benefit, I would clarify that lenalidomide maintenance is the SoC, while ixazomib may represent an alternative for patients in whom lenalidomide is not an option.

11) A press release just came out stating that the other co-primary endpoint of the cassiopeoa study (pfs from maintenance) was met, with a PFS hazard ratio (HR) of 0.53 (95% CI, 0.42-0.68) and a 47% reduction in the risk of disease progression or death with daratumumab. Might be worth to mention. This could become another SoC in this setting.

12) 2.22: when reporting the safety of Belantamab, I recommend rewording that Belamaf side effects are manageable, as 3/4 having ocular toxicity with 1/4 of G3-4 events is pretty concerning for the development of this drug, particularly when used in earlier lines when other options are available. My opinion is that we still need to work to improve this issue before a wide applicability of this compound.

Author Response

Dear Reviewer,  

We sincerely thank-you for your time in reading our manuscript entitled “Recent Advances on the Pathobiology and Treatment of Multiple Myeloma” and your invaluable suggestions, many of which we have incorporated into the revised version of the manuscript. We also appreciate your kind comments about the structure and overall content of this review.

In detail regarding your comments:

1] Additional information on impact/significance of t(14;16), amp(1q) and del(17p) has been incorporated into the respective sections.

2] Spelling error corrected.

3] Re-iterated the fact there is sufficient evidence that AutoSCT is safe and effective in elderly patients following improved patient selection measures and supportive care. Your suggested reference was indeed very appropriate and an interesting read.

4] EMNO2/HO95 study added to Table 2.

5] Agree with your statement that thalidomide is indeed still used and an acceptable alternative to lenalidomide in many European countries, text changed to reflect this fact.

6] Comments added to highlight that long-term follow up is anticipated to determine whether indeed upfront addition of anti-CD38 mAbs leads to improvement in PFS2/OS.

7] Valid argument; wording altered that despite the IMW frailty score being somewhat involved [and honestly in practice often neglected] it is a very valuable tool and should be assessed in all patients.

8] Detail on outcomes in patients >65yo in the SWOG0777 study added.

9] Additional paragraph added to discuss the potential utility and evidence behind tandem AutoSCT added with thanks as we had neglected to mention this treatment strategy.

10] Section slightly reworded to re-iterate that there is no OS benefit from ixazomib maintenance and that is a suitable alternative where lenalidomide is not an option/available.

11] Thank-you for bringing our attention to this press release and while this is interesting news, given it has not yet been presented or published in a peer-review journal, we have elected not include this information in this version of the manuscript.

12] The sentence alluding that the corneal toxicity with Belamaf is manageable has been removed altogether as the side effects, including significant corneal toxicity and the challenges this poses, is discussed further on in more detail.

Again, thank-you for your time and comments which are all very much appreciated.

Kind regards,

Slavisa Ninkovic and Hang Quach

Reviewer 2 Report

This is a well comprehensive review of the literature on myeloma pathbiology and therapy. Considering the literature, I believe it does not contribute much in terms of novelty and knowledge enhancement. The review is still well done. My major criticism concerns the lack in the biological part of the description of the mechanisms of evolution and clonal selection with the recent hypotheses of spatial and regional clonal heterogeneity. Furthermore, the part on the bone microenvironment is very minimal and not updated. Many important citations on the biology of myeloma-bone diseasew are missing.

Author Response

Dear Reviewer,

We thank-you for your time in reading our manuscript entitled “Recent Advances on the Pathobiology and Treatment of Multiple Myeloma” and your comments about this being a comprehensive review of the literature on myeloma pathobiology and therapy. While we agree there are no novel concepts or any original work presented, the manuscript serves the purpose of what it is, a review, providing the reader with comprehensive and up-to date information on the most important concepts regarding the pathobiology and approach to treatment of myeloma. We also agree that certain sections on the biology could be expanded, especially the non-immune tumour microenvironment, which deserves to be a review in its own right probably in something like a special edition of Cancers focusing on myeloma though we feel we touch on the major concepts and provide the reader with some additional useful references. A few additional insights on clonal evolution have been added in this version of the manuscript. Again we thank you sincerely for your time and efforts with our manuscript.

Kind regards,

Slavisa Ninkovic and Hang Quach